# Oncogenic Targets Regulated by Tumor-Suppressive *miR-30c-1-3p* and *miR-30c-2-3p*: *TRIP13* Facilitates Cancer Cell Aggressiveness in Breast Cancer

**DOI:** 10.3390/cancers15164189

**Published:** 2023-08-21

**Authors:** Reiko Mitsueda, Hiroko Toda, Yoshiaki Shinden, Kosuke Fukuda, Ryutaro Yasudome, Mayuko Kato, Naoko Kikkawa, Takao Ohtsuka, Akihiro Nakajo, Naohiko Seki

**Affiliations:** 1Department of Digestive Surgery, Breast and Thyroid Surgery, Graduate School of Medical and Dental Sciences, Kagoshima University, Kagoshima 890-8520, Japan; k7854142@kadai.jp (R.M.); k0848362@kadai.jp (H.T.); k4483271@kadai.jp (Y.S.); k3624054@kadai.jp (K.F.); k7682205@kadai.jp (R.Y.); takao-o@kufm.kagoshima-u.ac.jp (T.O.); k4814560@kadai.jp (A.N.); 2Department of Functional Genomics, Graduate School of Medicine, Chiba University, Chiba 260-8670, Japan; mayukokato@chiba-u.jp (M.K.); naoko-k@hospital.chiba-u.jp (N.K.)

**Keywords:** microRNA, breast cancer, *miR-30c-1-3p*, *miR-30c-2-3p*, *TRIP13*

## Abstract

**Simple Summary:**

Two passenger strand microRNAs (miRNAs), *miR-30c-1-3p* and *miR-30c-2-3p*, were identified as tumor-suppressive miRNAs in breast cancer (BrCa) cells. Seven genes (*TRIP13*, *CCNB1*, *RAD51*, *PSPH*, *CENPN*, *KPNA2*, and *MXRA5*) were putative targets of these miRNAs, and their expression was closely involved in BrCa molecular pathogenesis. Among these targets, inhibition of TRIP13 significantly suppressed aggressive phenotypes of BrCa cells.

**Abstract:**

Accumulating evidence suggests that the *miR-30* family act as critical players (tumor-suppressor or oncogenic) in a wide range of human cancers. Analysis of microRNA (miRNA) expression signatures and The Cancer Genome Atlas (TCGA) database revealed that that two passenger strand miRNAs, *miR-30c-1-3p* and *miR-30c-2-3p*, were downregulated in cancer tissues, and their low expression was closely associated with worse prognosis in patients with BrCa. Functional assays showed that *miR-30c-1-3p* and *miR-30c-2-3p* overexpression significantly inhibited cancer cell aggressiveness, suggesting these two miRNAs acted as tumor-suppressors in BrCa cells. Notably, involvement of passenger strands of miRNAs is a new concept of cancer research. Further analyses showed that seven genes (*TRIP13*, *CCNB1*, *RAD51*, *PSPH*, *CENPN*, *KPNA2*, and *MXRA5*) were putative targets of *miR-30c-1-3p* and *miR-30c-2-3p* in BrCa cells. Expression of seven genes were upregulated in BrCa tissues and predicted a worse prognosis of the patients. Among these genes, we focused on *TRIP13* and investigated the functional significance of this gene in BrCa cells. Luciferase reporter assays showed that *TRIP13* was directly regulated by these two miRNAs. *TRIP13* knockdown using siRNA attenuated BrCa cell aggressiveness. Inactivation of TRIP13 using a specific inhibitor prevented the malignant transformation of BrCa cells. Exploring the molecular networks controlled by miRNAs, including passenger strands, will facilitate the identification of diagnostic markers and therapeutic target molecules in BrCa.

## 1. Introduction

According to a report by the World Health Organization, breast cancer (BrCa) is the most common cancer among women worldwide; approximately 2.3 million women are diagnosed each year, with 700,000 BrCa-related deaths [1]. It is estimated that one in five women will develop BrCa during their lifetime [2]. The development of methods for the early diagnosis of BrCa and the discovery of new treatment regimens are important issues in BrCa research.

BrCa is a heterogeneous cancer and is classified into several subtypes according to histological and molecular classification [3,4,5]. For example, BrCa is classified as luminal, human epidermal growth factor receptor 2-enriched, and triple-negative types depending on the presence or absence of hormone receptors (estrogen and progesterone receptors) and epidermal growth factor receptor 2 in BrCa cells. Furthermore, the luminal type can be classified as luminal-A or luminal-B type according to the expression of Ki-67, a cell cycle marker [6]. Because the malignancy and outcomes of patients vary greatly among subtypes, treatment regimens also differ for each subtype [7,8]. In the future, further classification using molecular markers will facilitate the selection of different therapeutic regimens.

In the post-genome era, researchers have found that various noncoding RNAs are transcribed in cells and are deeply involved in critical functions, such as cell differentiation, proliferation, migration, and immune responses [9,10]. Among these noncoding RNAs, microRNAs (miRNAs) are short, single-stranded RNAs that act as fine controllers of gene expression depending on their sequences. Interestingly, a single miRNA can simultaneously control the expression of many genes [9,11]. Therefore, the presence or absence of miRNAs can disrupt the expression of target genes in cells and contribute to the malignant transformation of normal cells [12,13]. Many studies have revealed that aberrantly expressed miRNAs behave as oncogenes and tumor suppressors through targeting their corresponding genes in cancer cells [14].

Recently, we generated a BrCa miRNA expression signature by RNA sequencing and identified tumor-suppressive miRNAs and their target oncogenes in BrCa cells [15,16]. Our previous study revealed that *miR-101-5p* was downregulated in BrCa tissues, and its low expression predicted a worse prognosis in patients [16]. Ectopic expression assays showed that *miR-101-5p* attenuated BrCa malignant phenotypes by controlling several genes (e.g., *HMGB3*, *ESRP1*, *GINS1*, *TPD52*, *SRPK1*, *VANGL1*, and *MAGOHB*) whose expression levels are closely involved in the molecular pathogenesis of BrCa [16]. Importantly, *miR-101-5p* is annotated as a passenger strand miRNA derived from pre-*miR-101* in miRNA databases.

The general concept of miRNA biogenesis, two types of mature miRNAs are derived from pre-miRNAs. One strand (the guide strand) is selected for loading into the miRNA-Induced Silencing Complex (miRISC). The miRISC (including the guide strand) target mRNAs for silencing based on sequence depending manner. On the contrary, the passenger strand miRNAs (the other strand of pre-miRNA) are thought to be degraded in the cytoplasm and have previously been considered nonfunctional [17]. However, recent studies have shown that some passenger strands of miRNAs act as oncogenes or tumor suppressors in cancer cells [18,19]. Therefore, in cancer-miRNA research, it is essential to consider passenger strands derived from pre-miRNAs as well.

Analysis of our BrCa miRNA signature revealed that *miR-30c-1-3p* and *miR-30c-2-3p* (the passenger strands) were downregulated in cancer tissues. A large amount of cohort data obtained from The Cancer Genome Atlas (TCGA) database confirmed that *miR-30c-1-3p* and *miR-30c-2-3p* were downregulated in cancer tissues. Moreover, low expression of these targets was closely associated with worse prognosis in patients. Functional assays showed that these miRNAs exerted antitumor functions in BrCa cells by controlling several genes closely involved in BrCa malignant transformation, e.g., *TRIP13*, *CCNB1*, *RAD51*, *PSPH*, *CENPN*, *KPNA2*, and *MXRA5*.

In this study, we found that *miR-30c-3p* and its corresponding genes were involved in the malignant transformation of BrCa cells. These miRNA-targeted molecules may be candidates for the early diagnosis and treatment of BrCa.

## 2. Materials and Methods

### 2.1. Cell Lines and BrCa Clinical Specimens

Two BrCa Cell lines (MDA-MB-157 and MDA-MB-231) were used in this study. Both cell lines were obtained from Public Health England (Salisbury, UK).

The current study was conducted in accordance with the guidelines of the Declaration of Helsinki and was approved by the Ethics Committee of Kagoshima University (approval number 160038 28-65; date of approval: 19 March 2021).

### 2.2. Analysis of miRNAs and miRNA Target Genes in Patients with BrCa

The sequences of *miR-30* were confirmed using miRBase ver. 22.1 (https://www.mirbase.org, accessed on 10 July 2020) [20].

We used gene expression profiles of BrCa clinical specimens (GEO accession number: GSE118539) and TargetScanHuman ver.8.0 (https://www.targetscan.org/vert_80/ (accessed on 20 January 2023)) to search for genes regulated by *miR-30c-1-3p* and *miR-30c-2-3p* [21].

Expression data of the target genes from BrCa clinical tissues was obtained from GEPIA2 (http://gepia2.cancer-pku.cn/#index (accessed on 10 April 2023)) [22]. The clinical significance of genes in BrCa was obtained from OncoLnc (http://www.oncolnc.org/ (accessed on 10 April 2023)) [23,24,25].

### 2.3. Analysis of Molecular Pathways Using Gene Set Enrichment Analysis (GSEA) Software

We explored *TRIP13*-mediated molecular pathways by GSEA 4.3.2. From the TCGA-BRCA data, *TRIP13* expression levels were divided into high and low expression groups according to Z-score for BrCa patients. A ranked list of genes was created based on the log_2_ ratio comparing the expression level of each gene between the two groups. Gene ranking was performed by comparing the expression level of each gene between the two groups. Further analysis was performed by applying the Hallmark gene set from the Molecular Signatures Database [26,27,28].

### 2.4. RNA Extraction and Reverse Transcription Quantitative Polymerase Chain Reaction (RT-qPCR)

Total RNA from BrCa cell lines was isolated using Isogen II (NIPPON GENE Co., Ltd., Tokyo, Japan). cDNA was synthesized using High-Capacity cDNA Reverse Transcription Kit (catalog no.: 4368814, ThermoFisher Scientific Inc., Waltham, MA, USA). Gene expression was analyzed by real time PCR using SYBR green assay (ThermoFisher Scientific) on StepOnePlus Real-Time PCR System (Applied Biosystems, Foster City, CA, USA). An internal control in gene expression assays was β-Glucuronidases (GUSB). The sequences of primers for SYBR Green assays are summarized in Appendix A. The reagents used in this study were listed in Appendix A. The procedures for RNA extraction and qRT-PCR were described in our previous studies [29,30].

### 2.5. Transfection with Small Interfering RNA (siRNAs) and miRNAs

Opti-MEM (Gibco, Carlsbad, CA, USA) and LipofectamineTM RNAiMax Transfection Reagent (Invitrogen, Carlsbad, CA, USA) were used for transfection of small-sized RNA (siRNA and miRNA) into BrCa cell lines. The experimental protocol conforms to our previous studies [29,30]. The siRNAs and miRNAs used in this study are listed in Appendix A.

### 2.6. Cell Proliferation, Invasion and Migration Assays in BrCa Cells

Cell proliferation, invasion, and migration assays were performed in BrCa cells. Briefly, cell proliferation was assessed using XTT assays (Sigma-Aldrich, St. Louis, MO, USA); invasion was evaluated using Matrigel chamber assays with Corning BioCoat Matrigel (Corning, New York, NY, USA); and migration was examined using chamber assays with Corning BioCoat cell culture chambers (Corning). Details of the procedures are included in our previous studies [29,30].

### 2.7. Western Blotting and Immunohistochemistry

Western blotting and immunohistochemical analysis were performed according to our previous studies [29,30]. Anti-thyroid hormone receptor interactor 13 (TRIP13) human rabbit polyclonal IgG was used as a primary antibody. The antibodies used in the study are listed in Appendix A. A list of clinical specimens evaluated by immunohistochemistry is given in Appendix A.

### 2.8. RNA Immunoprecipitation (RIP) Assays

RIP assays were performed using a MagCapture microRNA Isolation Kit, Human Ago2 (FUJIFILM Wako Pure Chemical Corporation, Osaka, Japan) according to the manufacturer’s protocol. The expression level of *TRIP13* bound to Ago2-conjugated miRNAs was assessed using qRT-PCR.

### 2.9. Plasmid Construction and Dual-Luciferase Reporter Assays

Vector construction and dual-luciferase reporter assays were performed as described in our previous studies [19,31]. The vector insertion sequences are shown in Appendix A, and the reagents used are listed in Appendix A.

### 2.10. Statistical Analyses

Statistical analyses were performed using JMP Pro 16 (SAS Institute Inc., Cary, NC, USA). Differences between two groups were analyzed using Welch’s *t*-test, and those between multiple groups were analyzed using Dunnett’s test. Survival rates were analyzed by Kaplan–Meier survival curves and log-rank test.

## 3. Results

### 3.1. Expression and Clinical Significance of miR-30c-1-3p and miR-30c-2-3p in BrCa Clinical Specimens

Analysis of the miRBase database (release 22.1) showed that *miR-30c-1-3p* and *miR-30c-2-3p* were annotated as passenger strands of miRNAs and were derived from pre-*miR-30c-1* and pre-*miR-30c-2*, respectively. Seed sequences of these miRNAs were identical (Figure 1A). Human *miR-30c-1* was located on chromosome 1p34.2, whereas *miR-30c-2* was located on chromosome 6q13.

Analysis of our original miRNA expression signature created by RNA sequencing showed that some members of the *miR-30* family (*miR-30a*, *miR-30b*, *miR-30c*, *miR-30d*, and *miR-30e*) exhibited low expression in cancer tissues compared with normal tissues.

Accordingly, we next validated the expression levels of the *miR-30* family using TCGA datasets. The expression levels of *miR-30c-1-3p* and *miR-30c-2-3p* were significantly reduced in BrCa tissues (Figure 1B). Moreover, low expression of these miRNAs was associated with a significantly poor prognosis (based on the 10-year survival rate) compared with high expression of these miRNAs (Figure 1C). The expression levels of the two miRNAs were compared across patient subtypes. The expression level of *miR-30c-1-3p* was higher in TNBC than in luminal. *miR-30c-2-3p* showed lower expression in TNBC compared to luminal (Appendix A). Examination of prognosis by patient subtype showed that in luminal patients, patients with low *miR-30c-1-3p* expression had a poor prognosis (Appendix A). The *miR-30c-5p* (the guide strand) was also analyzed in the same way. Expression of *miR-30c-5p* was reduced in BrCa tissues compared with normal tissues. However, *miR-30c-5p* expression did not affect the prognosis of BrCa patients (Appendix A).

Our recent studies revealed that some passenger strands of miRNAs were closely involved in the molecular pathogenesis of human cancers. In this study, we focused on *miR-30c-1-3p* and *miR-30c-2-3p* and investigated the functional significance of these miRNAs with the aim of identifying their target genes in BrCa cells.

### 3.2. Tumor-Suppressive Roles of miR-30c-1-3p and miR-30c-2-3p in BrCa Cells

The tumor-suppressive roles of *miR-30c-1-3p* and *miR-30c-2-3p* were evaluated by ectopic expression of *miR-30c-1-3p* and *miR-30c-2-3p* in two triple-negative BrCa cell lines, MDA-MB-157 and MDA-MB-231. 

Transient transfection of *miR-30c-1-3p* and *miR-30c-2-3p* inhibited BrCa cell proliferation (Figure 2A). Cancer cell invasion and migration abilities were markedly inhibited by *miR-30c-1-3p* and *miR-30c-2-3p* expression in MDA-MB-157 and MDA-MB-231 cells (Figure 2B,C). Typical images of BrCa cells during invasion and migration assays after *miR-30c-1-3p* and *miR-30c-2-3p* transfection are shown in Appendix A.

Based on our current analysis, two miRNAs (*miR-30c-1-3p* and *miR-30c-2-3p*) showed tumor-suppressive roles through targeting several oncogenic genes in BrCa cells.

### 3.3. Identification of Genes Controlled by miR-30c-1-3p and miR-30c-2-3p in BrCa Cells

To detect genes that were controlled by *miR-30c-1-3p* and *miR-30c-2-3p* in BrCa cells, we carried out in silico database analysis and combined these findings with our gene expression data. Our strategy for miRNA target searching is shown in Figure 3.

The TargetScan Human database (release 8.0) revealed that a total of 3,154 genes had *miR-30c-1-3p* and *miR-30c-2-3p* binding sites in their 3′ untranslated regions (UTRs). Using BrCa clinical specimens, a gene expression profile was obtained (Gene Expression Omnibus accession number: GSE118539); 525 genes were identified as upregulated (log_2_ fold ratio > 2.0) in cancer tissues compared with normal tissues. By merging the two datasets, 62 genes were selected as *miR-30c-1-3p* and *miR-30c-2-3p* targets in BrCa cells (Table 1).

**Figure 3 cancers-15-04189-f003:**
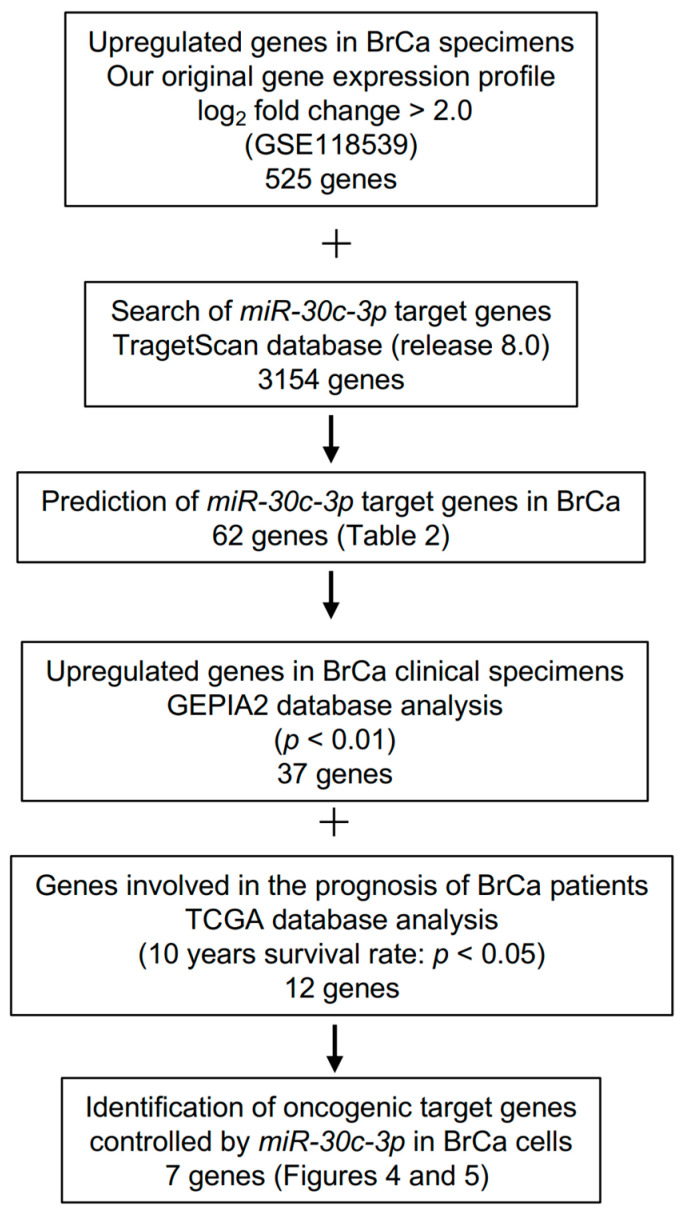
Strategy for identification of *miR-30c-1-3p* and *miR-30c-2-3p* targets in BrCa cells. To identify target genes, we used our original mRNA profile (GEO accession number: GSE118539) and the TargetScan Human database (release 8.0). By merging the two sets of data, we identified 62 genes as *miR-30c-1-3p* and *miR-30c-2-3p* targets. Among 62 targets, we narrowed down the clinically significant genes in BrCa using two databases, GEPIA2 (http://gepia2.cancer-pku.cn/#analysis (accessed on 10 April 2023) and OncoLnc (http://www.oncolnc.org/ (accessed on 10 April 2023). Finally, seven oncogenic genes were selected as *miR-30c-1-3p* and *miR-30c-2-3p* targets in BrCa cells.

### 3.4. Clinical Significance of Putative Target Genes of miR-30c-1-3p and miR-30c-2-3p in BrCa

Moreover, the 62 selected genes were subjected to clinicopathological analysis using the TCGA-BRCA dataset. Among these genes, 37 genes were significantly overexpressed in BrCa tissues (*n* = 1085) compared with normal tissues (*n* = 291; *p* < 0.01). In addition, 12 genes showed statistically significant correlations with poor overall survival (based on the 10-year survival rate, *p* < 0.05).

We finally selected seven genes (*TRIP13*, *CCNB1*, *RAD51*, *PSPH*, *CENPN*, *KPNA2*, and *MXRA5*) as targets of *miR-30c-1-3p* and *miR-30c-2-3p* in BrCa cells. Our results showed that these genes were upregulated in BrCa tissues (Figure 4), and their high expression significantly predicted poor prognosis (10-year overall survival) in patients with BrCa (Figure 5).

### 3.5. Clinical Significance of TRIP13 in BrCa 

Among these targets, we focused on *TRIP13* and performed further analyses of its function in BrCa cells. Recent studies have shown that TRIP13 is a key regulator of meiotic recombination and the spindle assembly checkpoint [32,33]. Exploitation of the checkpoint inhibition process may have applications in the treatment of BrCa.

Immunohistochemistry was performed to analyze TRIP13 expression in BrCa clinical specimens. TRIP13 protein was highly expressed in cancer lesions but weakly expressed in noncancerous areas (Figure 6). Tissue information is shown in Appendix A.

A multivariate Cox proportional hazards model showed that high expression of *TRIP13* was an independent prognostic factor for overall survival after adjusting for well-known clinical prognostic factors (age, T stage, N stage, and M stage; Figure 7A).

### 3.6. TRIP13-Mediated Molecular Pathways in BrCa Cells

To investigate TRIP13-mediated molecular pathways in BrCa cells, we performed GSEA using TCGA-BRCA RNA-sequencing data. “E2F targets”, “G_2_M checkpoint”, and “MYC target” pathways were enriched in patients showing high *TRIP13* expression compared with those in patients showing low expression (Table 2, Figure 7B).

### 3.7. Direct Regulation of TRIP13 by miR-30c-1-3p and miR-30c-2-3p in BrCa Cells

First, we investigated whether the expression of *TRIP13* was controlled by *miR-30c-1-3p* and *miR-30c-2-3p* in BrCa cells. The expression levels of *TRIP13* mRNA and protein were markedly suppressed by ectopic expression of *miR-30c-1-3p* and *miR-30c-2-3p* in BrCa cells (Figure 8A,B). The full western blotting image is shown in Appendix A.

To confirm the incorporation of *TRIP13* mRNA into the RNA-induced silencing complex (RISC) in BrCa cells, RIP assays were conducted (Figure 8C). Ago2-bound miRNAs and mRNAs were isolated via immunoprecipitation of Ago2, which plays a central role in the RISC. qRT-PCR using immunoprecipitation-isolated samples demonstrated significantly higher levels of *TRIP13* mRNA in *miR-30c-1-3p* and *miR-30c-2-3p* transfected cells compared with control cells. These findings provided evidence of the significant incorporation of *TRIP13* into the RISC. 

To investigate whether *miR-30c-1-3p* and *miR-30c-2-3p* bound directly to the 3′ UTR of *TRIP13* in BrCa cells, dual-luciferase reporter assays were conducted (Figure 9A). TargetScan database analysis revealed that *miR-30c-1-3p* and *miR-30c-2-3p* shared a binding site. The luminescence intensity was significantly decreased following co-transfection with *miR-30c-1-3p* or *miR-30c-2-3p* and a vector containing the *miR-30c-3p* binding site in the 3′ UTR of *TRIP13* (Figure 9B). Conversely, co-transfection of a construct without the *miR-30c-3p* binding site (deleted *miR-30c-3p* binding site) showed no decrease in the luminescence intensity (Figure 9B). These data indicated that *miR-30c-1-3p* and *miR-30c-2-3p* bound directly to *TRIP13* and regulated *TRIP13* expression in BrCa cells.

### 3.8. Effect of TRIP13 siRNA and the TRIP13 Inhibitor DCZ0415 on TRIP13 Function in BrCa Cell

Next, to analyze the oncogenic roles of *TRIP13* in MDA-MB-231 BrCa cells, we performed knockdown assays using siRNAs targeting *TRIP13*. Two types of siRNAs (si*TRIP13*-1 and si*TRIP13*-2) markedly suppressed *TRIP13* expression at both the mRNA and protein levels in BrCa cells (Figure 10A,B). Full western blotting images are shown in Appendix A.

Cell proliferation assays showed that si*TRIP13*-transfected MDA-MB-231 cells show significantly reduced cell growth (Figure 10C).

In an analysis using a TRIP13 inhibitor (DCZ0415), cell proliferation was suppressed in a concentration-dependent manner (Figure 10D).

## 4. Discussion

Our previous studies demonstrated that *miR-99a-3p* (the passenger strand of pre-*miR-99a*) and *miR-101-5p* (the passenger strand of pre-*miR-101*) functioned as tumor-suppressive miRNAs in BrCa cells. Their target genes *FAM64A* and *GINS1* were shown to be aberrantly expressed in BrCa tissues, and their expression levels were closely associated with the molecular pathogenesis of BrCa [16,30]. Such new findings demonstrating that the passenger strands of miRNAs derived from pre-miRNAs are involved in cancer pathogenesis indicate that passenger strands of miRNAs should be analyzed alongside guide strands.

Based on the BrCa miRNA signature, we focused on *miR-30c-1-3p* and *miR-30c-2-3p* and demonstrated that these miRNAs behaved as tumor-suppressive miRNAs by controlling several oncogenic genes in BrCa cells. The *miR-30* family consists of six miRNAs (*miR-30a*, *miR-30b*, *miR-30c-1*, *miR-30c-2*, *miR-30d*, and *miR-30e*), each of which generates guide and passenger strands from their respective precursors. Interestingly, the six miRNAs are clustered in pairs on three chromosomes (*miR-30e*/*miR-30c-1* on chromosome 1p34.2, *miR-30c-2*/*miR-30a* on chromosome 6q13, and *miR-30b*/*miR-30d* on chromosome 8q24.22). The seed sequences of *miR-30c-1-3p* and *miR-30c-2-3p* that control the target genes are identical (UGGGAG).

Previous studies have shown the downregulation of *miR-30c-5p* in several types of cancers, including breast cancer, and the oncogenes it regulates have been implicated in various cancer pathways, such as cell proliferation, metastasis, and drug resistance. On the contrary, *miR-30c-3p* (the passenger strand) has not been reported in detail [34,35,36,37]. A large number of cohort analysis by TCGA database revealed that low expression levels of *miR-30c-5p* did not affect the prognosis of BrCa patients. In contrast, BrCa patients with low miRNAs (*miR-30c-1-3p* and *miR-30c-2-3p*) expressions had clear impact on prognosis. Therefore, this study focused on two types of miRNAs, *miR-30c-1-3p* and *miR-30c-2-3p*. In the estrogen receptor-negative BrCa subtype, nuclear factor kappa B (NF-κB) signaling is frequently activated. Additionally, *miR-30c-2-3p* has been shown to act as a negative regulator of NF-κB signaling, and ectopic expression of *miR-30c-2-3p* attenuates cell proliferation by targeting TRADD and *CCNE1* in BrCa cells [38]. In another study, overexpression of the circular RNA *circ0072995* was shown to promote the invasion and migration of cancer cells by adsorbing *miR-30c-2-3p* in MDA-MB-231 cells [39]. These reports are consistent with our current findings and strongly indicated that *miR-30c-2-3p* acted as a tumor-suppressive miRNA in BrCa cells.

Some reports have described the roles of *miR-30c-1-3p* and *miR-30c-2-3p* in other cancer types. For example, in lung adenocarcinoma, the long noncoding RNA *LINC00346* was shown to adsorb *miR-30c-2-3p* and abolish its tumor-suppressive function. Overexpression of *LINC00346* promotes the development of lung adenocarcinoma through regulation of the *miR-30c-2-3p*/cell cycle signaling pathway [40]. N6-methyladenosine (m6A) is the most common modification in the mammalian RNA transcriptome and is broadly present in mRNAs and certain noncoding RNAs [41]. Recent studies have suggested that alterations in m6A modification patterns are deeply involved in tumorigenesis [42]. Methyltransferase-like 14 (*METTL14*) is a key RNA methyltransferase involved in m6A modification. A recent study showed that *METTL14* enhances the maturation of *miR-30c-1-3p* and that *miR-30c-1-3p* expression inhibits lung cancer malignant transformation [43]. Moreover, *METTL14*-mediated m6A modification has also been reported to be involved in *miR-30c-2-3p* regulation in gastric cancer [44]. Aberrant expression of genes involved in m6A modification and regulation of miRNAs in BrCa cells will be important research topics in the future.

A feature of miRNAs is that the target genes they control differ depending on the type of cancer. In this study, we attempted to search for genes regulated by tumor-suppressive *miR-30c-1-3p* and *miR-30c-2-3p* in BrCa cells. In total, seven genes (*TRIP13*, *CCNB1*, *RAD51*, *PSPH*, *CENPN*, *KPNA2*, and *MXRA5*) were identified as putative targets of *miR-30c-1-3p* and *miR-30c-2-3p*, and their expression levels were found to be closely associated with poor prognosis in patients. As a future study, it will be necessary to analyze miRNAs and target molecules controlled by miRNAs by subtype of BrCa patients.

Based on these findings, we focused on *TRIP13*, a member of the large superfamily of AAA+ ATPase proteins [45]. The AAA+ ATPase family is involved in a wide range of biological processes, including protein folding and DNA recombination, replication, and repair [46,47]. Our data showed that aberrant expression of *TRIP13* was deeply involved in the malignant transformation of BrCa cells. Notably, *TRIP13* has been shown to be overexpressed in several types of cancers, and its aberrant expression is involved in the malignant transformation of various types of cancer cells, including BrCa cells [48,49]. In lung cancer cells, *TRIP13* knockdown inhibited malignant phenotypes, e.g., increased apoptosis, induced cell cycle arrest, and inhibited the proliferation, invasion, and migration abilities. Furthermore, overexpression of *TRIP13* was associated with tumor metastasis through activation epithelial-mesenchymal transformation pathways [48]. TRIP13 is a novel mitotic checkpoint-silencing protein. Overexpression of *TRIP13* is a hallmark of cancer cells exhibiting chromosomal instability, especially in certain BrCa with poor prognosis [49]. In head and neck cancer, overexpressed *TRIP13* interacts with the DNA-protein kinase C complex and activates the DNA repair process, thereby affecting drug resistance [50]. A recent study demonstrated that the TRIP13 inhibitor DCZ0415 impairs nonhomologous end-joining repair and attenuates cancer cell growth in hepatocellular carcinoma [51]. Furthermore, combining DCZ0415 and olaparib (a poly [ADP-ribose] polymerase [PARP1] inhibitor) has synergistic anticancer effects against hepatocellular carcinoma cells [51]. PARP1 inhibitors and CDK4/6 inhibitors have also been used in the treatment of BrCa. Combining these drugs with TRIP13 inhibitors may lead to synergistic anticancer effects, thereby facilitating the development of new treatment regimens.

## 5. Conclusions

In this study, TCGA analysis revealed that low expression levels of *miR-30c-1-3p* and *miR-30c-2-3p* adversely affected the prognosis of patients with BrCa. Ectopic expression of these miRNAs attenuated the malignant phenotypes of BrCa cells, suggesting that these miRNAs acted as tumor-suppressive miRNAs in BrCa cells. In total, seven genes (*TRIP13*, *CCNB1*, *RAD51*, *PSPH*, *CENPN*, *KPNA2*, and *MXRA5*) were putative targets of *miR-30c-1-3p* and *miR-30c-2-3p*, and their high expression levels were associated with a worse prognosis in patients. *TRIP13* was directly regulated by *miR-30c-1-3p* and *miR-30c-2-3p*, and its overexpression facilitated BrCa cell aggressiveness. Based on the tumor-suppressive miRNAs analysis, it was possible to identify genes that were closely related to the molecular pathogenesis of BrCa.

## Figures and Tables

**Figure 1 cancers-15-04189-f001:**
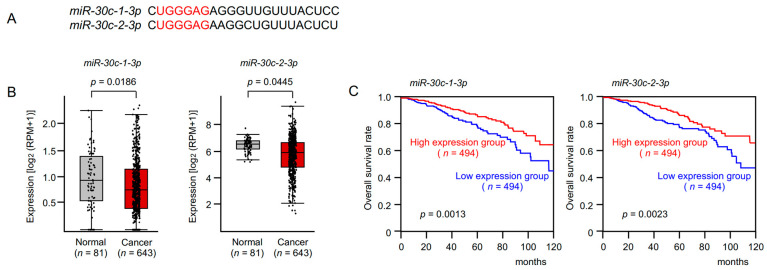
Expression and clinical significance of *miR-30c-1-3p* and *miR-30c-2-3p* in BrCa clinical specimens. (**A**) The mature sequences of *miR-30c-1-3p* and *miR-30c-2-3p* are given. Seed sequences of these miRNAs are shown in red. (**B**) Expression levels of *miR-30c-1-3p* and *miR-30c-2-3p* were evaluated by TCGA-BRCA database analysis. In total, 643 BrCa tissues and 81 normal epithelial tissues were analyzed. (**C**) Kaplan–Meier survival curve analyses of patients with BrCa using data from TCGA-BRCA dataset. The patients were divided into high and low expression groups according to their miRNA expression (based on median expression). The red line shows the high expression group, and the blue line shows the low expression group.

**Figure 2 cancers-15-04189-f002:**
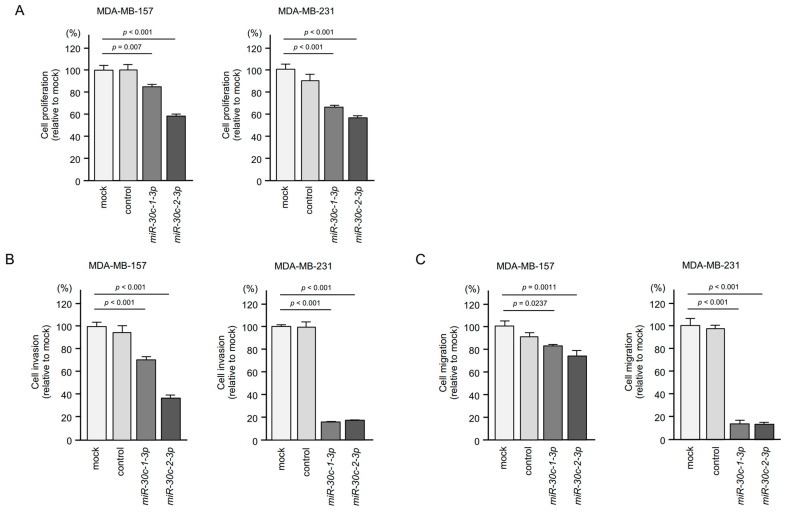
Functional assays of *miR-30c-1-3p* and *miR-30c-2-3p* in BrCa cell lines (MDA-MB-157 and MDA-MB-231). (**A**) Cell proliferation was assessed using XTT assays 72 h after miRNA transfection. (**B**) Cell invasion was assessed using Matrigel invasion assays at 48 h after seeding miRNA-transfected cells into the chambers. (**C**) Cell migration was assessed using a membrane culture system at 48 h after seeding miRNA-transfected cells into the chambers.

**Figure 4 cancers-15-04189-f004:**
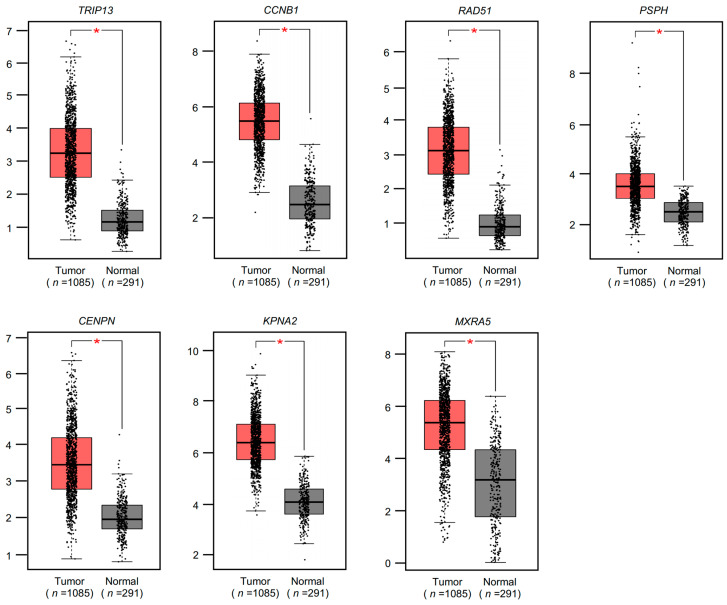
Expression levels of seven targets in BrCa specimens. Analysis of the expression levels of seven genes (*TRIP13*, *CCNB1*, *RAD51*, *PSPH*, *CENPN*, *KPNA2*, and *MXRA5*) in BrCa clinical specimens using TCGA-BRCA datasets. All these genes were upregulated in BrCa tissues (*n* = 1085) compared with normal tissues (*n =* 291) (* *p* < 0.01).

**Figure 5 cancers-15-04189-f005:**
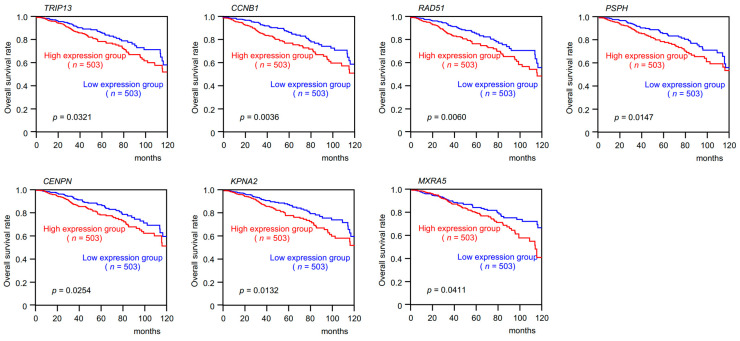
Clinical significance of seven targets in BrCa specimens. Kaplan–Meier curves of 10-year overall survival rates according to the expression levels of the seven target genes (*TRIP13*, *CCNB1*, *RAD51*, *PSPH*, *CENPN*, *KPNA2*, and *MXRA5)*. The patients (*n* = 1006) were divided into high and low expression groups according to the median gene expression level. The red lines represent the high expression group, and the blue lines represent the low expression group. High expression levels of these genes were significantly correlated with poor prognosis in patients with BrCa.

**Figure 6 cancers-15-04189-f006:**
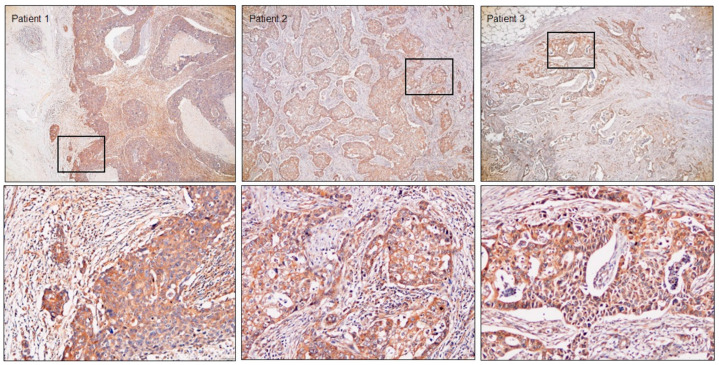
Expression of TRIP13 in BrCa tissues. Immunohistochemical staining of TRIP13 was confined to cancer tissues, whereas weak staining was observed in the noncancerous area. Magnification: 40× (**upper**) and 200× (**lower**).

**Figure 7 cancers-15-04189-f007:**
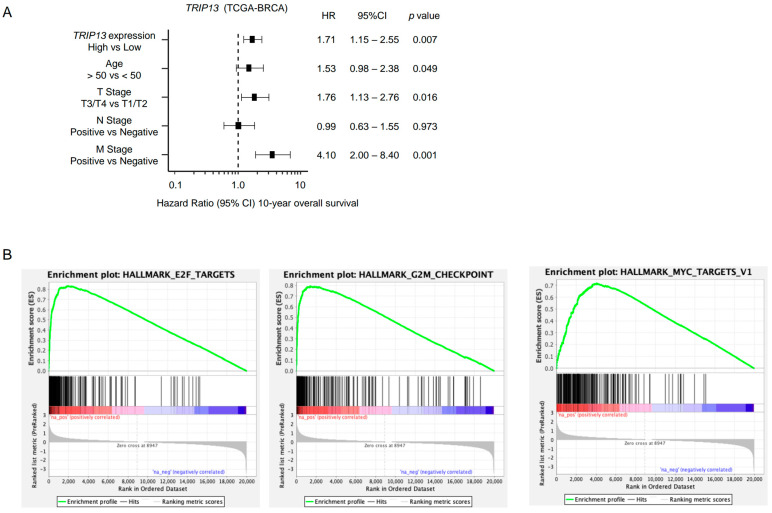
Clinical significance of *TRIP13* in BrCa specimens. (**A**) Forest plot showing multivariate Cox proportional hazards regression analysis of 10-year overall survival. Patients with high *TRIP13* expression showed significantly low overall survival. The data were obtained from TCGA-BRCA datasets. (**B**) TRIP13-mediated pathways identified by gene set enrichment analysis. The top three enrichment plots (E2F targets, G_2_M checkpoint, and MYC targets) are presented in the high *TRIP13* expression group for patients with BrCa.

**Figure 8 cancers-15-04189-f008:**
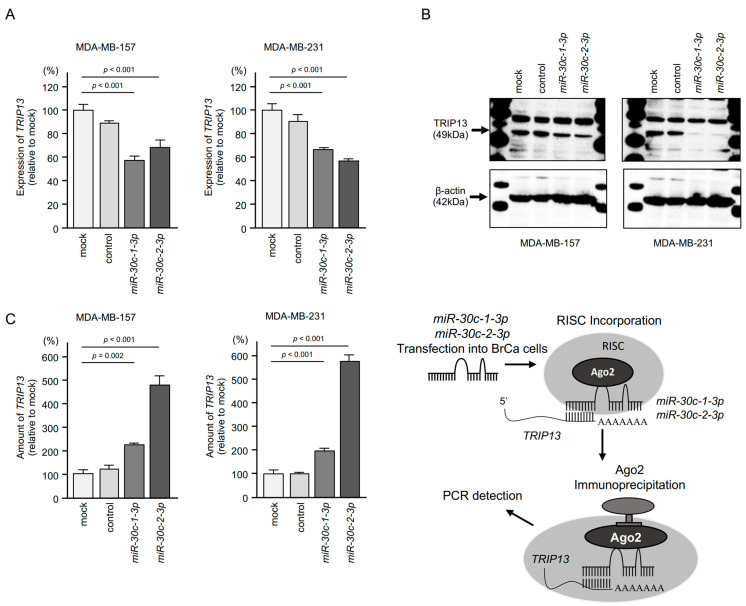
Control of *TRIP13* expression by *miR-30c-1-3p* and *miR-30c-2-3p* in BrCa cells. (**A**) qRT-PCR analyses demonstrating the downregulation of *TRIP13* mRNA expression at 72 h after *miR-30c-1-3p* and *miR-30c-2-3p* transfection into BrCa cells. *GUSB* was used as an internal control. (**B**) Western blot analyses showing significantly reduced expression of TRIP13 protein 72 h after *miR-30c-1-3p* and *miR-30c-2-3p* transfection into BrCa cells. β-actin was used as an internal loading control. (**C**) Isolation of RISC-incorporated *TRIP13* mRNA by Ago2 immunoprecipitation. Direct *TRIP13* expression by *miR-30c-1-3p* and *miR-30c-2-3p* in BrCa cells is demonstrated. Schematic illustration of the RIP assay is shown in the right. qRT-PCR suggested that *TRIP13* mRNA was significantly incorporated into the RISC.

**Figure 9 cancers-15-04189-f009:**
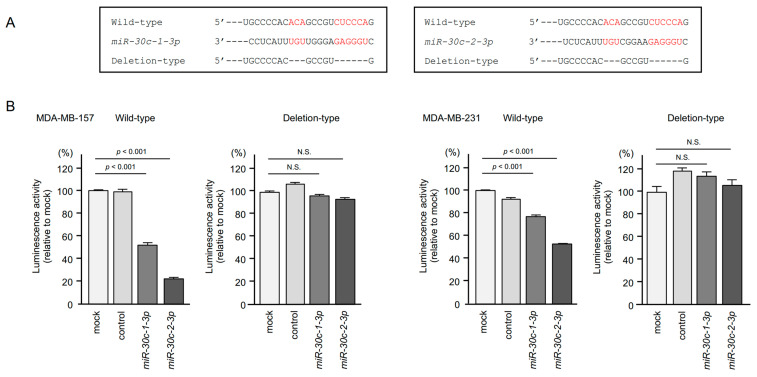
*miR-30c-1-3p* and *miR-30c-2-3p* bound directly to the 3′ UTR of *TRIP13* in BrCa cells. (**A**) TargetScan database analysis predicting putative *miR-30c-3p*-binding sites in the 3′ UTR of *TRIP13*. The sequence of the binding sites is highlighted in red. (**B**) In dual luciferase reporter assays, co-transfection of *miR-30c-1-3p* or *miR-30c-2-3p*, and a vector containing the *miR-30c-3p* binding site in the 3′ UTR of *TRIP13* showed decreased luminescence activity in BrCa cells (N.S.: not significant compared with the mock group).

**Figure 10 cancers-15-04189-f010:**
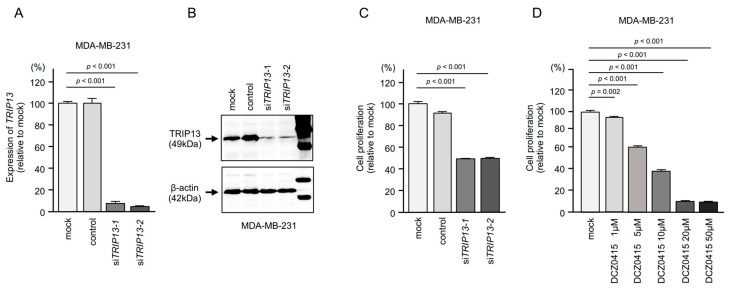
Oncogenic function of *TRIP13* in BrCa cells. (**A**) The inhibitory effects of two different siRNAs targeting *TRIP13* (si*TRIP13*-1 and si*TRIP13*-2) expression were examined. *TRIP13* mRNA levels were effectively inhibited by each siRNA. (**B**) TRIP13 protein levels were effectively inhibited by each siRNA (si*TRIP13*-1 and si*TRIP13*-2). (**C**) Cell proliferation was assessed using XTT assays 72 h after siRNA transfection into MDA-MB-231 cells. Cell proliferation ability was significantly reduced after knockdown of *TRIP13*. (**D**) MDA-MB-231 cells were treated with DCZ0415, a TRIP13 inhibitor. Cell proliferation ability was significantly blocked by DCZ0415 in a concentration-dependent manner.

**Table 1 cancers-15-04189-t001:** Candidate gene targets of *miR-30c-1-3p/miR-30c-2-3p* significantly upregulated in our breast cancer mRNA profile.

Entrez Gene ID	Gene Symbol	Gene Name	*miR-30c-1-3p/* *miR-30c-2-3p* Total Binding Sites	mRNA Profile log_2_ Fold Change	Gene Expression *p* Value	10-Year Overall Survival *p* Value
10024	*TROAP*	Trophinin associated protein	1	4.59	<0.01	0.256
9319	*TRIP13*	Thyroid hormone receptor interactor 13	1	3.72	<0.01	0.032
83461	*CDCA3*	Cell division cycle associated 3	1	3.62	<0.01	NA ^1^
8968	*HIST1H3F*	Histone cluster 1, H3f	1	3.50	>0.01	0.733
9123	*SLC16A3*	Solute carrier family 16 (monocarboxylate transporter), member 3	2	3.38	<0.01	0.300
113730	*KLHDC7B*	Kelch domain containing 7B	1	3.35	>0.01	0.300
952	*CD38*	CD38 molecule	1	3.31	>0.01	0.068
891	*CCNB1*	Cyclin B1	3	3.24	<0.01	0.004
147841	*SPC24*	SPC24, NDC80 kinetochore complex component	2	3.23	<0.01	0.532
1281	*COL3A1*	Collagen, type III, alpha 1	1	3.20	<0.01	0.310
92359	*CRB3*	Crumbs family member 3	1	3.19	<0.01	0.531
5888	*RAD51*	RAD51 recombinase	3	3.07	<0.01	0.006
8638	*OASL*	2’-5’-oligoadenylate synthetase-like	1	3.05	<0.01	0.596
6664	*SOX11*	SRY (sex determining region Y)-box 11	1	3.03	>0.01	0.027
4085	*MAD2L1*	MAD2 mitotic arrest deficient-like 1 (yeast)	1	2.99	<0.01	0.058
5723	*PSPH*	Phosphoserine phosphatase	1	2.98	<0.01	0.015
84900	*RNFT2*	Ring finger protein, transmembrane 2	2	2.95	<0.01	0.623
3017	*HIST1H2BD*	Histone cluster 1, H2bd	3	2.89	<0.01	0.244
6772	*STAT1*	Signal transducer and activator of transcription 1, 91kDa	1	2.88	<0.01	0.810
2537	*IFI6*	Interferon, alpha-inducible protein 6	1	2.83	<0.01	0.813
1462	*VCAN*	Versican	1	2.80	<0.01	0.095
317754	*POTED*	POTE ankyrin domain family, member D	6	2.78	>0.01	NA
2065	*ERBB3*	erb-b2 receptor tyrosine kinase 3	2	2.66	<0.01	0.131
57156	*TMEM63C*	Transmembrane protein 63C	1	2.64	<0.01	0.725
10051	*SMC4*	Structural maintenance of chromosomes 4	1	2.59	<0.01	0.271
79814	*AGMAT*	Agmatine ureohydrolase (agmatinase)	1	2.58	>0.01	0.050
55423	*SIRPG*	Signal-regulatory protein gamma	2	2.57	>0.01	0.021
4261	*CIITA*	Slass II, major histocompatibility complex, transactivator	7	2.56	>0.01	0.017
2151	*F2RL2*	Coagulation factor II (thrombin) receptor-like 2	4	2.51	<0.01	0.970
8534	*CHST1*	Carbohydrate (keratan sulfate Gal-6) sulfotransferase 1	2	2.50	>0.01	0.052
154467	*CCDC167*	Coiled-coil domain containing 167	1	2.48	<0.01	0.479
4939	*OAS2*	2’-5’-oligoadenylate synthetase 2, 69/71kDa	2	2.48	<0.01	0.506
55839	*CENPN*	Centromere protein N	1	2.48	<0.01	0.025
22797	*TFEC*	Transcription factor EC	1	2.46	>0.01	0.731
8477	*GPR65*	G protein-coupled receptor 65	2	2.41	>0.01	0.315
921	*CD5*	CD5 molecule	1	2.37	>0.01	0.001
554313	*HIST2H4B*	Histone cluster 2, H4b	1	2.37	<0.01	NA
1951	*CELSR3*	Cadherin, EGF LAG seven-pass G-type receptor 3	2	2.37	>0.01	0.719
4582	*MUC1*	Mucin 1, cell surface associated	1	2.36	<0.01	0.905
4860	*PNP*	Purine nucleoside phosphorylase	1	2.32	>0.01	0.110
55824	*PAG1*	Phosphoprotein membrane anchor with glycosphingolipid microdomains 1	3	2.32	>0.01	0.197
3838	*KPNA2*	Karyopherin alpha 2 (RAG cohort 1, importin alpha 1)	1	2.32	<0.01	0.013
1122	*CHML*	Choroideremia-like (Rab escort protein 2)	2	2.31	>0.01	0.170
7371	*UCK2*	Uridine-cytidine kinase 2	1	2.28	>0.01	0.062
1734	*DIO2*	Deiodinase, iodothyronine, type II	3	2.28	<0.01	0.411
653269	*POTEI*	POTE ankyrin domain family, member I	3	2.26	>0.01	NA
22996	*TTC39A*	Tetratricopeptide repeat domain 39A	2	2.24	<0.01	0.189
9735	*KNTC1*	Kinetochore associated 1	1	2.22	>0.01	0.173
9603	*NFE2L3*	Nuclear factor, erythroid 2-like 3	3	2.22	>0.01	0.972
3070	*HELLS*	Helicase, lymphoid-specific	1	2.21	<0.01	0.327
8038	*ADAM12*	ADAM metallopeptidase domain 12	1	2.17	<0.01	0.344
25878	*MXRA5*	Matrix-remodelling associated 5	1	2.17	<0.01	0.041
27338	*UBE2S*	Ubiquitin-conjugating enzyme E2S	1	2.16	<0.01	0.800
55248	*TMEM206*	Transmembrane protein 206	2	2.15	<0.01	0.214
11006	*LILRB4*	Leukocyte immunoglobulin-like receptor, subfamily B (with TM and ITIM domains), member 4	2	2.12	<0.01	0.246
150372	*NFAM1*	NFAT activating protein with ITAM motif 1	3	2.07	>0.01	0.304
83481	*EPPK1*	Epiplakin 1	2	2.06	<0.01	0.341
201254	*STRA13*	Stimulated by retinoic acid 13	1	2.04	>0.01	0.487
2187	*FANCB*	Fanconi anemia, complementation group B	1	2.02	>0.01	0.017
4495	*MT1G*	Metallothionein 1G	1	2.01	>0.01	0.442
8270	*LAGE3*	L antigen family, member 3	2	2.01	<0.01	0.862
10962	*MLLT11*	Myeloid/lymphoid or mixed-lineage leukemia (trithorax homolog, Drosophila); translocated to, 11	1	2.01	>0.01	0.602

^1^ Not Available.

**Table 2 cancers-15-04189-t002:** TRIP13-mediated pathways identified by gene set enrichment analysis (GSEA).

Name	Normalized Enrichment Score	FDR *q*-Value
HALLMARK_E2F_TARGETS	3.561	*q* < 0.001
HALLMARK_G2M_CHECKPOINT	3.395	*q* < 0.001
HALLMARK_MYC_TARGETS_V1	3.038	*q* < 0.001
HALLMARK_MYC_TARGETS_V2	2.602	*q* < 0.001
HALLMARK_MTORC1_SIGNALING	2.477	*q* < 0.001
HALLMARK_MITOTIC_SPINDLE	2.473	*q* < 0.001
HALLMARK_UNFOLDED_PROTEIN_RESPONSE	1.975	*q* < 0.001
HALLMARK_SPERMATOGENESIS	1.906	*q* < 0.001
HALLMARK_DNA_REPAIR	1.691	0.003
HALLMARK_INTERFERON_ALPHA_RESPONSE	1.655	0.003

## Data Availability

The data presented in this study are available on request from the corresponding author.

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
