# Peer review of "Oncogenic Targets Regulated by Tumor-Suppressive miR-30c-1-3p and miR-30c-2-3p: TRIP13 Facilitates Cancer Cell Aggressiveness in Breast Cancer"

_cancers, 2023, doi:10.3390/cancers15164189_

Round 1
Reviewer 1 Report
In general, the organization of the manuscript is difficult to follow, a mix of clinic and basic experiments should be changed. The supplementary files are not showed.
In particular, please delate reference 9 and 11 from line 63 because you used the same cite in line 64.
The methods should be briefly described and the plasmids, miRNAs and siRNAs sequences should be added.
Discuss the difference between invasion and migration in cells transfected with miR-30c-2-3p (Figure 2). If you have a more potent effect in invasion than in migration, why you decide to analyzed a target involved in DNA reparation instead of invasion?
Figure 6 should be labeled.
In Figure 9 discuss Ì´20% of inhibition in luciferase in cells MDA-MB-231 transfected with miR-30c-1-3p. Considering that you have 40% and 99% of inhibition of mRNA and TRIP13 protein expression, respectively.
Discuss the difference between inhibition of proliferation in cells MDA-MB-231 transfected with siRNA (Figure 10) and miRNA (Figure 2), that it is almost 60% and 40%, respectively. Considering that that siRNA inhibits near 90% of TRIP13 protein expression (Figure 10B) versus near 99% of TRIP13 protein expression inhibition with miRNA (Figure 8B).
Reviewer 2 Report
The authors focused the passenger strands of miR-30c (miR-30c-1-3p and miR-30c-2-3p) in regulating human breast cancer. Since inhibition of breast cancer by miR-30c has been widely reported, major concerns have to be addressed before further consideration.
1) How about the role of the guide strands of miR-30c in breast cancer? Do the guide strands synergy with passenger strands of miR-30c? The expression levels and expression change of guide strands and passenger strands of miR-30c should be analyzed and compared in breast cancer.
2) Is the tumor suppressing function of miR-30c-1-3p and miR-30c-2-3p tumor subtype specific?
3) The role of miR-30c-1-3p/miR-30c-2-3-TRIP13 signaling in regulating breast tumor growth in vivo (for example, animal model) should be determined.
minor revision of English is required.
Round 2
Reviewer 2 Report
I still think the expression pattern and their correlation with survival of miR-30c-1-3p and miR-30c-2-3p in different subtype of breast cancer should be determined, at least using public database.
Author Response
Comment:
I still think the expression pattern and their correlation with survival of miR-30c-1-3p and miR-30c-2-3p in different subtype of breast cancer should be determined, at least using public database.
Response:
Following the reviewers' comments, microRNAs (miR-30c-1-3p and miR-30c-2-3p) expression were analyzed by patient subtype.
The analysis results are shown in Figure S2 and S3. I also added the following sentences to the Results 3.1 (indicated by a red marker).
The expression levels of the two miRNAs were compared across patient subtypes. The expression level of miR-30c-1-3p was higher in TNBC than in Luminal. miR-30c-2-3p showed lower expression in TNBC compared to Luminal (Figure S2). Examination of prognosis by patient subtype showed that in Luminal patients, patients with low miR-30c-1-3p expression had a poor prognosis (Figure S3).